# MAGNET 🧲: Improving the Multilingual Fairness of Language Models with Adaptive Gradient-Based Tokenization

**Orevaoghene Ahia[1]   Sachin Kumar[2,3]   Hila Gonen[1]   Valentin Hofmann[2]**
**Tomasz Limisiewicz[4]   Yulia Tsvetkov[1]   Noah A. Smith[1,2]**
[1]University of Washington   [2]Allen Institute for AI   [3]The Ohio State University
[4]Charles University
oahia@cs.washington.edu

## Abstract

In multilingual settings, non-Latin scripts and low-resource languages are usually disadvantaged in terms of language models' utility, efficiency, and cost. Specifically, previous studies have reported multiple modeling biases that the current tokenization algorithms introduce to non-Latin script languages, the main one being over-segmentation. In this work, we propose MAGNET—multilingual adaptive gradient-based tokenization—to reduce over-segmentation via adaptive gradient-based subword tokenization. MAGNET learns to predict segment boundaries between byte tokens in a sequence via sub-modules within the model, which act as internal boundary predictors (tokenizers). Previous gradient-based tokenization methods aimed for uniform compression across sequences by integrating a single boundary predictor during training and optimizing it end-to-end through stochastic reparameterization alongside the next token prediction objective. However, this approach still results in over-segmentation for non-Latin script languages in multilingual settings. In contrast, MAGNET offers a customizable architecture where byte-level sequences are routed through language-script-specific predictors, each optimized for its respective language script. This modularity enforces equitable segmentation granularity across different language scripts compared to previous methods. Through extensive experiments, we demonstrate that in addition to reducing segmentation disparities, MAGNET also enables faster language modelling and improves downstream utility.

## 1   Introduction

Despite the proliferation of generative language models (LMs) in English, their non-English counterparts are far from being widely adopted. While multilingual LMs offer several advantages such as resource efficiency and cross-lingual generalization, the performance disparities across languages remain a significant challenge. Previous work has largely attributed these disparities to training data imbalances across languages [43, 34, 29, 24]. Recent work, however, highlights that *tokenization*— the way input text is segmented—can considerably degrade not only model performance but also training and inference costs on account of overly fragmenting certain languages and scripts [3, 33]. Subword segmentation algorithms used to build LM tokenizers [28, 39, 22, 40] typically segment the training corpus relying on frequency statistics alone. Due to data imbalances, they obtain high compression in high-resource languages, while majority of languages are over-fragmented. This issue disproportionately affects non-Latin scripts covering languages spoken by billions of people, which are not only less frequent in such corpora, but can require up to $4\times$ more bytes to represent the same information.

38th Conference on Neural Information Processing Systems (NeurIPS 2024).

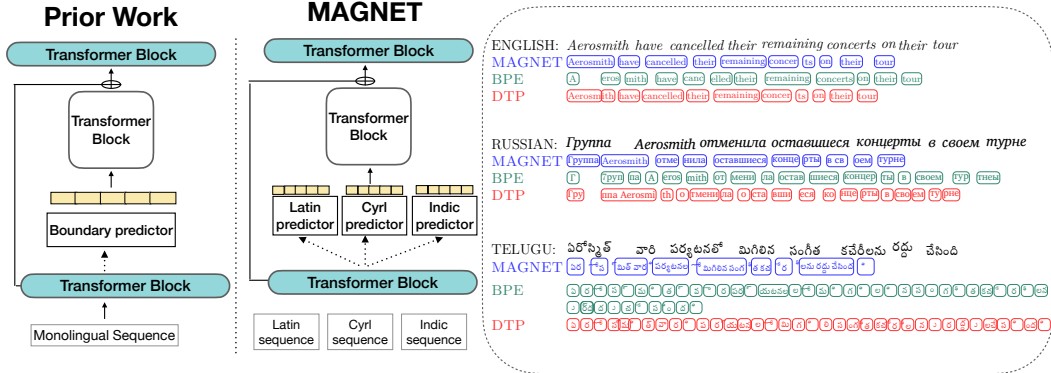

Figure 1: MAGNET routes byte-level sequences via language-script specific boundary predictors. These predictors infer boundaries leading to equitable segmentation across languages. Prior work infers boundaries with a single predictor across languages and leads to over-segmentation.

To address these challenges, prior work has instead proposed to build tokenizer-free models by directly training on character or byte sequences [45, 4]. Operating on smaller or finer-grained segments leads to significantly higher compute- and memory-intense modeling, caused by much longer sequences. To alleviate this issue, recent work introduced a tokenization "layer" within the model architecture by pooling fixed or dynamic length contiguous character representations into smaller sets of patch representations [30, 9, 41, 31, 14, 15], resulting in models optimized with a sequence "compression rate" in mind. While these can improve efficiency, they are mostly suited for character-level modeling for scripts whose characters can be mapped to single Unicode codepoint. Due to the extensive number of codepoints in Unicode, character-based vocabularies can be extremely large for multilingual models. Since many of those characters may never appear during training, "out-of-vocabulary" issues similar to those experienced with subword models may arise if we only included codepoints in the training data [28].[1] When extending these methods to training byte-level multilingual models, we observe that the disparities in fragmentation rates across languages persist . For instance, the English text "Fellow wrestlers also paid tribute to Luna." and its Telugu equivalent, "తోటి మల్ల యుద్ధకారులు కూడా లూనాకు నివాళులు అర్పించారు.", contain 43 and 148 UTF-8 bytes, respectively. A fixed compression ratio for both languages will result in the Telugu text getting over fragmented with requiring close to $3\times$ more tokens than English.

In this work, we propose MAGNET (multilingual adaptive gradient-based tokenization) to reduce this disparity in tokenizer-free multilingual LMs. Our goal is to obtain end-to-end multilingual language modeling with gradient-based subword tokenization that results in high and similar sequence compression across languages with varying scripts. We leverage hourglass transformers [30, 31] to efficiently route byte-level sequences through language-script specific internal boundary predictors trained to infer word boundaries between byte tokens in a sequence. These boundary predictors are trained end-to-end through stochastic reparameterisation [20, 27]. The inferred boundaries are then used to pool representations of contiguous tokens in the same segment, after which the pooled (narrow) representation is passed into the rest of the transformer block. Unlike previous gradient-based tokenization approaches that apply the same compression rate to all languages in the pretraining data by incorporating a single boundary predictor within their model architectures, MAGNET employs modularity. It incorporates multiple language-script specific predictors to achieve equitable segmentation granularity across different language scripts. We test the effectiveness of MAGNET on equitable fragmentation, model efficiency, and downstream task performance across nine typologically different languages, comparing to byte-level models without compression and existing gradient-based tokenization models [31]. Our extensive experiments demonstrate that our approach MAGNET results in more equitable tokenization when compared to subword-tokenizers, byte-level tokenizers and previous gradient-based tokenization models. This in turn leads to faster modelling and competitive performance across downstream-tasks.[2]

---

[1]Another disparity with character-level tokenizers is that Chinese-Japanese-Korean scripts use a high number of Unicode codepoints.

[2]Code and data are publicly available at https://github.com/orevaahia/magnet-tokenization

## 2  MAGNET: Multilingual Adaptive GradieNt-basEd Tokenization

Our goal is to build a multilingual byte-level language model with equitable segmentation across languages. We propose to achieve this by dedicating a separate module within the model for each writing script, to serve as an internal tokenizer for languages that use that script. Our proposed model, called MAGNET, builds on *hourglass transformers* [30, 31], an architecture that was introduced to efficiently handle long sequences in tokenizer-free models. We make several simple but important modifications to this architecture in order to obtain equitable segmentation across languages, while maintaining a high quality of multilingual modeling. In what follows, we explain the main concepts of hourglass transformers, and then introduce the modifications we make to accommodate equitable multilingual modeling.

### 2.1  Background: Hourglass Transformers

The hourglass transformer [31, 30] is a hierarchical architecture for efficiently handling long sequences. The architecture has three main components, each consisting of one or more transformer layers: A **tokenization submodule** which takes as input a byte sequence and outputs a segmentation, a **language modeling submodule** that takes as input the predicted segments or tokens and is then trained to perform next token prediction, and an **upsampling module** that takes as input the hidden representations of the segmentations and converts them back to a byte sequence on which a typical language modeling loss can be applied. Considering this model as a blackbox, it still performs byte-level language modeling, however, it requires significantly less compute thanks to the tokenization submodule.

**Gradient-based Tokenization**   This submodule performs two steps. First, the given input sequence $x_1, \ldots, x_N$ (where each $x_t$ is a byte in our case) is encoded using a small transformer network (with causal attention) to produce a sequence of hidden vectors $h_1^T, \ldots, h_N^T$. Next, a *boundary predictor* takes as input each $h_t$ and predicts a scalar value between 0 and 1, indicating the probability of position $t$ to be the end of a segment. It is implemented as

$$\hat{b}_t = p(b_t = 1) = \sigma(\text{MLP}_\phi(h_t)), \tag{1}$$

where MLP indicates a multi-layer perceptron and $\sigma$ is the sigmoid function. To convert the soft probabilities to hard segment predictions, a Bernoulli distribution is sampled from, defined by $\hat{b}_t$. Since the sampling operation will make the process non-differentiable, hard Gumbel-sigmoid is used, a stochastic reparameterization of the Bernoulli distribution, following Nawrot et al. [31]:

$$b_t = \text{sigmoid}\left[ \log \frac{\hat{b}_t u}{(1 - \hat{b}_t)(1 - u)}^{\frac{1}{\tau}} \right], \quad u \sim \text{Uniform}(0, 1) \tag{2}$$

where $\tau$ is a hyper-parameter. Since this module is differentiable, the segmentations are learned during training of the full model.[3] This module is referred to as "gradient-based tokenization."

**Language Modeling**   Given a sequence of segment boundaries $b_t \in \{0, 1\}$ from the boundary predictor, this submodule first pools the hidden states belonging to the same segment by averaging them to form a sequence of representations $h_1^P, \ldots, h_k^P$.[4] Let $t_1, \ldots, t_k$ indicate the positions at which a boundary is sampled, i.e., for any contiguous pair $t_j, t_{j+1}$, the sequence $x_{t_j+1} \ldots x_{t_{j+1}}$ forms a "token" ending at position $t_{j+1}$.[5] The input representation of this "token" is defined as $hP_j = \frac{1}{t_{j+1}-t_j} \sum_{t=t_j+1}^{t_{j+1}} hT_t$ . These representations are then passed through the middle block of transformer layers (with causal attention) to obtain another sequence of hidden representations $h_1^M, \ldots, h_k^M$. From the perspective of a subword-tokenizaton based language model, this module is equivalent to the transformer blocks without the input and output embedding layers.

**Upsampling**   This module converts $h_l^M$ to probabilities over a byte vocabulary. This involves, first, upsampling the output of the middle block to the original resolution by duplication followed by skip

---

[3]Nawrot et al. [31] also explore learning the segmentations using supervision from predefined word or subword boundaries. However, it is not a viable solution for all languages and does not resolve the unfairness issues.

[4]$P$, $M$, and $T$ denote representations in the middle transformer block, after pooling and at the token level.

[5]The first token is defined as $x_0 \ldots x_{t_1}$ and the last token as $x_{t_k+1} \ldots x_N$.

connection: $h_t^U = h_{\lceil \frac{t-k+1}{k} \rceil}^M + h_t^T$.[6] These vectors are further passed through a small transformer network followed by an unembedding layer and a softmax to get a probability distribution over which language modeling loss (cross entropy) can be computed. To prevent the boundary predictor from collapsing and trivially predicting each position $t$ as a boundary, Nawrot et al. [31] propose adding a regularizer to the LM objective: $-\log \text{Binomial}(\beta; l, k)$ where,

$$\text{Binomial}(\beta; N, k) = \binom{N}{k} \beta^k (1-\beta)^{N-k}, \quad \text{and} \quad k = \sum_N b_t. \tag{3}$$

Here $\beta \in [0, 1]$ is a hyperparameter, $k$ defines the number of predicted segments or tokens. Intuitively, this loss nudges the model to find a $k$ close to $\beta l$ which is the mode of the Binomial distribution. In other words, $\beta$ allows to control the compression rate of the input sequence to approximately $\frac{1}{\beta} \times$. Setting it as 1 will lead to every position being predicted as boundary whereas setting it to 0 will cause no boundaries to be predicted.

## 2.2 Adaptive Gradient-Based Tokenization via Multiple Boundary Predictors

To encode the same information, different languages require different number of bytes, owing to their different inherent efficiencies [3, 25, 33] as well as restrictions imposed by Unicode mappings, where non-Latin languages (e.g., Indian languages) may require up to 4 bytes per character.[78] In multilingual models, setting the same compression rate (via $\beta$) for all languages will lead to text in some languages getting segmented into much longer sequences,[9] see Equation (3). This disparity contributes to higher compute and memory costs for such languages as well as poorer model performance for downstream tasks [3]. This issue parallels subword tokenizers where languages with higher segmentation rates get disadvantaged due to longer context-length requirements to perform the same task and poorer in-context learning performance since the same context length fits fewer training instances than other languages leading to unfairness [5, 3, 25, 33].

To make tokenization more equitable, we propose MAGNET, which efficiently learns to segment sequences across languages and language scripts with similar granularity. As part of creating equitable segmentation, we aim to efficiently maximize sequence compression, without having a negative impact on downstream performance across languages.

**Introducing multiple gradient-based tokenizers**    To achieve this, we propose a modification to the model architecture that enables the processing of multiple language scripts. Each script has its own boundary predictor trained with distinct Binomial priors $\beta$ determined based on the scripts' Unicode encoding and also tailored to a desired compression rate. This allows us to achieve similar fragmentation rates across languages, due to variations in compression. The input sequence is tagged with its script[10] and we infer the segmentation by routing it through the appropriate boundary predictor. The remainder of the model architecture remains the same.

**Determining $\beta$ for equitable tokenizaton**    We use the binomial priors $\beta$ for each boundary predictor to control the rates of the resulting segmentations for the different scripts. Since we want to impose equitable lengths across languages, we set the different $\beta$ according to the following process. First, we choose an *anchor language* $L$ for each script in our training corpus and define a quantity *byte-to-word ratio* $\bar{R}$ for this script as follows. Let $\mathcal{X} = \{\mathbf{x}_1, \ldots, \mathbf{x}_D\}$ be a sample of text sequences in language $L$ from our training corpus with $|\mathbf{x}_i|$ denoting the byte-length and $\text{count}_{\text{words}}(\mathbf{x}_i)$ the number of words[11] in sequence $\mathbf{x}_i$. We define the average byte-to-word ratio $\bar{R}$ over $\mathcal{X}$ as:

$$\bar{R} = \frac{1}{D} \sum_{i=1}^{D} \frac{|\mathbf{x}_i|}{\text{count}_{\text{words}}(\mathbf{x}_i)} \tag{4}$$

---

[6]The hidden vectors are shifted by one in order to perform next token prediction.

[7]https://en.wikipedia.org/wiki/UTF-8

[8]Modeling characters directly may alleviate this issue; however, character-level multilingual models can explode vocabulary sizes.

[9]As is the case generally in multilingual modeling, also without boundary predictors.

[10]Determining the script of given text sequence is trivial, we assume every sequence contains a single script.

[11]For the purposes of this work, words are defined by whitespace boundaries.

We then set the prior $\beta_S$ for the corresponding script $S$ to be $1/\bar{R}$. Our final training objective over a single instance **x** is as follows:

$$\sum_{i=1}^{N} -\log p_\theta(x_i|x_{<i}) - \lambda \sum_S \mathbb{I}(\text{script}(\mathbf{x}) = S) \log \text{Binomial}(\beta_S; N, k)$$

where $\mathbb{I}$ is the indicator function and $\text{script}(\cdot)$ is a function assigning a writing script to a sequence of bytes **x**, such assignment can be easily obtained based on codepoint definitions in Unicode.

## 3  Experimental Setup

### 3.1  Language Modeling

**Pretraining Data**  We pretrain all models on nine languages (English, Spanish, French, Russian, Ukranian, Belarusian, Telugu, Bengali and Hindi) divided into three groups, written with distinct scripts: Latin, Cyrillic, and Indic (Brahmic) scripts. Our choice of selection is based on the linguistic diversity of these languages and the availability of data for downstream evaluation. Our pretraining data is obtained from the OSCAR dataset [32]. We present the statistics for each language in Appendix C.

**Baselines**  We compare MAGNET against Dynamic Token Pooling [31], which infers boundaries with a fixed binomial prior $\beta$ for every sequence, irrespective of the language script. This model is referred to as DTP in the rest of the paper. DTP has a single boundary predictor; we train two versions of this baseline with the binomial prior $\beta$ as 0.2 and 0.1 respectively yielding $5\times$ and $10\times$ compression respectively. We also compare against a byte-level decoder language model. To ensure fair model comparisons, this model has a similar architecture as DTP, but without any sequence compression.

**MAGNET configurations**  We compute the byte-word-ratios', choosing English, Russian and Telugu as anchor languages for each the language script, based on initial explorations. The FLO-RES [16] dataset is used for this purpose and the resulting ratios are approximately $5\times$, $10\times$, and $20\times$ for English, Russian, and Telugu, respectively. Based on these ratios, we train five MAGNET models with different binomial prior combinations maintaining the ratio but adjusting the multipliers. First, to optimize for word-level boundary segmentation, we use the original byte-to-word ratio configuration, i.e., $5\times$ compression for Latin, $10\times$ compression for Cyrillic and $20\times$ compression for Indic languages within the same model. The second configuration; $(1, 2, 4)\times$ is the average bytes-to-character ratio for English, Russian and Telugu. Hence, using this configura-

Table 1: Binomial prior choice for each MAGNET and DTP model configuration. These combinations of binomial priors determine the compression rate of sequences per language script. While DTP uses fixed priors for all languages, MAGNET is dynamic and script-specific.

| Configuration | Binomial Prior | | |
| | Latin | Cyrillic | Indic |
| --- | --- | --- | --- |
| DTP $5\times$ | 0.2 | 0.2 | 0.2 |
| DTP $10\times$ | 0.1 | 0.1 | 0.1 |
| MAGNET $(1, 2, 4)\times$ | 1 | 0.5 | 0.25 |
| MAGNET $(3, 6, 12)\times$ | 0.33 | 0.17 | 0.083 |
| MAGNET $(5, 10, 13)\times$ | 0.2 | 0.10 | 0.076 |
| MAGNET $(5, 10, 15)\times$ | 0.2 | 0.10 | 0.066 |
| MAGNET $(5, 10, 20)\times$ | 0.2 | 0.10 | 0.05 |

tion optimizes for fair byte-level-modelling with character-level granularity. The third configuration $(3, 6, 12)\times$ is based on a hypothesis that it would lead to fair subword-segmentation boundaries. Finally, since we apply a very high compression on Indic languages, we empirically test two additional configurations $(5, 10, 13)\times$ and $(5, 10, 15)\times$ with a reduced compression rate for Indic languages.

**Subword Tokenizer Training**  To compare the segmentation derived from MAGNET to traditional subword tokenizers, we create byte-level byte pair encoding (BPE) vocabularies containing 50K, 100K and 250K subword units on our pretraining data. We employ $\alpha$-sampling to train the tokenizers, typically used to improve representation of low-resource languages [10]. That is, we sample documents for each language according to a multinomial distribution with probabilities $\{q_i\}_{i=1...N}$, where: $q_i = \frac{p_i^\alpha}{\sum_{j=1}^{N} p_j^\alpha}$ with $p_i = \frac{n_i}{\sum_{k=1}^{N} n_k}$. This increases tokens for low-resource languages and has been shown to reduce bias towards high-resource ones. We consider $\alpha = 0.5, 0.3$ consistent with [10, 12].

**Downstream Datasets** To demonstrate the effectiveness of MAGNET, we evaluate by finetuning our trained models on several question answering and classification tasks. Specifically, we evaluate on XQuAD (question answering) [6], XNLI (natural language inference) [11], PAWS-X (paraphrase detection) [46] from XTREME [19], and SIB 200 (the topic classification) [2]. We provide a detailed language coverage across all tasks in Table 5. In addition, to test how adaptation capabilities of MAGNET, we evaluate on dialectal tasks, specifically ILI [48] the Indo-Aryan Language Identification (ILI) shared task and HaSCoSVa-2022 [7], hatespeech detection on Spanish dialects. We provide finetuning details in Appendix D.

## 3.2 Analyzing segmentation across models

The objective of this analysis is to compare segmentation granularity across different approaches. That is, we measure whether the same amount of information is conveyed through similar token counts across various languages. Following previous work [33, 3], we conduct this analysis with the parallel corpus FLORES-200 [16] focusing on the nine languages in our pretraining data.

For byte-level models, segmentation is done by converting each sentence to raw UTF-8 bytes and computing the average number of bytes per sentence. With the subword tokenizer, each sentence is segmented using the tokenizer we trained in 3.1, and the average number of resulting tokens computed across all sentences. As for gradient-based methods like DTP and our MAGNET, we feed each sentence into the model and retrieve a sequence of boundary predictions from the boundary predictor layers. The count of positive predictions determines the number of tokens per sentence.

## 4 Results

The goal of MAGNET is to learn equitable segmentation during training while maintaining high quality downstream performance. Ideally, we expect that MAGNET results in higher compression for the non-Latin script languages, hence balancing segmentation granularity across all languages. This, in turn, should improve modeling efficiency by reducing computational costs at training and inference time.

### 4.1 MAGNET results in equitable segmentation across language scripts.

We analyze the segmentation granularity, contrasting our method with byte-level, subword tokenization, and DTP as described in §3.2. Our results in Figure 2 show that MAGNET models produce similar segmentation rates for all languages. The improvement is particularly noticeable in non-Latin script languages that are most susceptible to over-segmentation with the baselines.

First, we compare byte-level segmentation to MAG-NET with the $(1, 2, 4)\times$ segmentation configuration. As described in §3.1, Indic languages have approximately four byte code-points to one character, Cyrillic languages have approximately two, and many Latin languages have a one-to-one mapping. Therefore, we expect that training MAGNET with the $(1, 2, 4)\times$ configuration will result in equitable byte-level modeling across all of these languages. Appendix Figure 7a shows that MAGNET $(1, 2, 4)\times$ results in a $3\times$ drop in the average number of tokens for the Indic languages, and close to $2\times$ drop for Cyrillic, while the Latin languages are not affected. Next, we compare segmentation between DTP $5\times$, MAGNET at $(5, 10, 13)\times$,

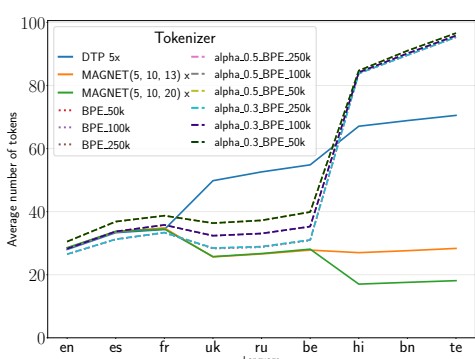

Figure 2: Average number of tokens after segmenting the FLORES dataset. Subword tokenizers and DTP result in over-segmentation in non-Latin script languages, while MAGNET closes the gap.

MAGNET at $(5, 10, 20)\times$ and subword-tokenizers at vocabulary sizes 50k, 100k and 250k with and without alpha sampling (see §3.1). We find that MAGNET models result in the most equitable segmentation across all languages. In fact, we measure a drop close to $5\times$ in the average number of tokens for the Indic languages compared to DTP and the subword tokenizers. Notably, we also see that a subword tokenizer with a large vocabulary size is required to achieve a lower segmentation rate on Cyrillic

languages, whereas for Indic languages, even with a large vocabulary and $\alpha$ sampling, we observe a pronounced disparity. This contrasts with findings from previous work that $\alpha$ sampling alleviates tokenization disparities [10]. Overall, these results suggest that MAGNET learns equitable segmentation across languages with diverse scripts, while fixed segmentation models like DTP and byte-based subword tokenizers are sub-optimal and very likely to result in over-segmentation.

## 4.2 MAGNET maintains performance on downstream tasks.

Our goal is to enforce equitable segmentation while maintaining model performance across tasks. In Table 2, we present results for the best-performing MAGNET model compared to DTP and byte-level models (we provide comparisons to all MAGNET models in §5.1). Overall, We find that MAGNET models perform better than DTP but are competitive with the byte-level models while being considerably faster and requiring less compute. MAGNET$(3, 6, 12)\times$ performs best on PAWS-X and SIB, while $(1, 2, 4)\times$ performs best on XNLI and XQUAD.

Figure 3: Language-specific accuracy on the XNLI task across byte-level, and all DTP and MAGNET models. Results are mostly competitive between byte-level and MAGNET models on Latin languages and Russian.

We report language-specific results on the XNLI dataset in Figure 3. We see better results with the MAGNET models even in some cases where the models in comparison optimize for a similar segmentation. For instance, on Spanish, MAG-NET at $(5, 10, 15)\times$ outperforms DTP at $5\times$. On Indic languages, MAGNET models are generally competitive with the DTP models. We provide more analysis on the trade-offs between downstream performance and segmentation in §5.1. Results on dialectal tasks are reported in §2 We see competitive results across all models on all the tasks, suggesting that there is also no negative impact as a result of adapting the models to their respective dialects.

Table 2: The average performance (accuracy) on downstream tasks on all languages across different models. We present results for the best-performing MAGNET model: $((3, 6, 12)\times$ for PAWS-X and SIB), $((1, 2, 4)\times$ for XQUAD and XNLI). Bold indicates the best overall performance. Table 6 provides more detailed language-specific results.

| Model | XNLI | PAWS-X | SIB | XQUAD | | Hascova | ILI |
|---|---|---|---|---|---|---|---|
| Byte-level | 68.68 | 82.18 | 71.05 | **44.62** | | 87.38 | 89.24 |
| DTP5$\times$ | 68.16 | 81.29 | 69.17 | 43.31 | | 86.87 | 89.17 |
| DTP10$\times$ | 67.31 | 75.99 | 67.83 | 35.83 | | **87.62** | 88.72 |
| MAGNET | **68.74** | **85.41** | **71.43** | 44.61 | | 87.25 | **89.27** |

| (a) In-language tasks. | (b) Dialectal tasks. |
|---|---|

## 4.3 MAGNET results in more efficient models.

Comparing inference times across all models, we expect that models which optimize for fixed compression like DTP would be only efficient for Latin languages because of their lower byte-to-word ratio. Hence, we anticipate that our routing strategy with the MAGNET models would result in an efficiency gain for non-Latin script languages. In Figure 4, we plot the inference time per language in XQUAD, relative to the inference time of the byte-level models. We show that MAGNET has a shorter inference time than the byte-level models, comparable to DTP for English and Russian and slightly lower for Hindi and Russian. If we assume the optimal compression rates for English and Russian to be $5\times$ and $10\times$, respectively, using a DTP model with a fixed compression rate for both languages requires training two separate monolingual models to obtain the ideal compression rate for each. However, training a

single MAGNET $(5, 10, 20)\times$ model that dynamically achieves $5\times$ compression rate for English and $10\times$ compression rate for Russian results in a lower inference time for both.

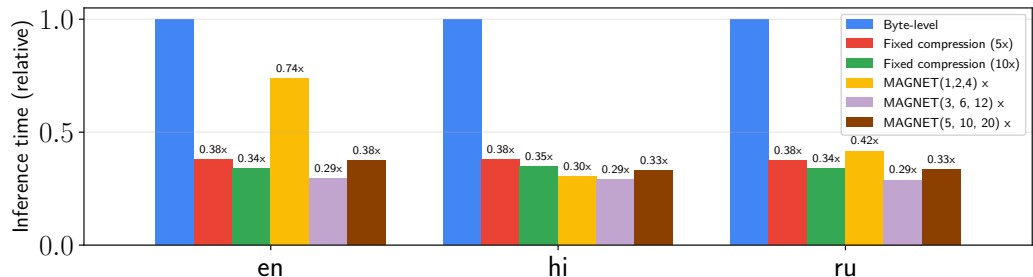

Figure 4: Inference time per language in XQUAD, relative to the byte-level model. MAGNET's inference time is shorter than the byte-level model and comparable to DTP for most of the languages.

## 5 Analysis and Discussion

### 5.1 Trade-off between downstream performance and equitable segmentation

Previous studies have reported correlations between compression and model performance [14]. We empirically investigate these tradeoffs by comparing downstream performance across different MAGNET configurations defined in Table 1. In the results reported in Table 3, we find that the configurations that perform best are MAGNET$(1, 2, 4)\times$ and MAGNET$(3, 6, 12)\times$. These configurations are equivalent to fair byte-level and subword modelling across all languages. We also report the average task performance per script at various compression rates in Figure 5. Here we are not looking to compare performance across language scripts, but rather to assess performance across different compression rates within each language script. Our results show that there is little to no drop in performance for Latin and

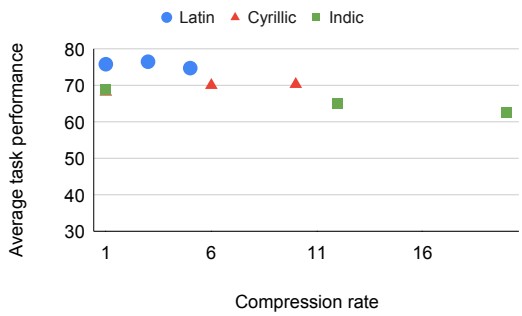

Figure 5: Average task performance vs compression trade off across language scripts.

Cyrillic languages as compression increases moderately. However, for Indic languages, we see an average of 5% drop in performance as compression increases.

Table 3: Results (accuracy) from ablations across all MAGNET configurations.

| Model | XNLI | PAWSX | SIB | XQUAD | Hascova | ILI |
|---|---|---|---|---|---|---|
| MAGNET$(1, 2, 4)\times$ | **68.74** | 83.30 | 71.02 | **44.61** | 86.75 | 88.62 |
| MAGNET $(3, 6, 12)\times$ | 68.35 | **85.41** | **71.43** | 43.72 | 87.13 | 88.57 |
| MAGNET $(5, 10, 20)\times$ | 67.50 | 80.44 | 71.13 | 41.06 | 87.25 | **89.27** |
| MAGNET $(5, 10, 13)\times$ | 66.28 | 79.85 | 70.64 | 41.92 | 87.88 | 88.35 |
| MAGNET $(5, 10, 15)\times$ | 67.96 | 84.79 | 68.79 | 42.81 | **88.37** | 88.67 |

| (a) In-language tasks | (b) Dialectal tasks |
|---|---|

### 5.2 What is the granularity of segmentation across different compression rates?

In §3.1, we highlight that the binomial prior is essential for determining the granularity of the segmentation derived from the boundary predictor. To intrinsically validate that MAGNET indeed learns

segmentations of similar lengths across different languages, we manually analyze examples from the SIB corpus, comparing them with DTP at 5× and 10×. As shown in Table 4 DTP at 5× produces word-level segmentation for all Latin languages while producing subword-level segmentation for Cyrillic and Indic languages. At 10×, we see word-level segmentation for Cyrllic languages, phrase-level segmentation for Latin languages and a mix of subword and word-level segmentation on Indic languages. To achieve word-level segmentation for all languages, DTP requires training three separate models. However MAGNET alleviates this requirement by producing a similar segmentation granularity across all the languages. For comparison, the segmentation granularity of the BPE tokenizer is highly sub-optimal for Indic languages as shown in Appendix Table 4. While the BPE tokenizer produces word-level segmentations for Latin and Cyrillic languages, it produces character-level segmentation for Indic languages. MAGNET, on the other hand, finds a good balance of segmentation granularity across languages.

Table 4: Segmentation of English, Spanish, Hindi, Russian, Ukranian and Telugu examples from the SIB corpus. While other models produce different segmentation granularity across languages, MAGNET consistently produces similar segmentation granularity across languages.

| Lang | Text | DTP 5x | DTP 10x | MAGNET$(5, 10, 20)\times$ | BPE 100k |
|---|---|---|---|---|---|
| en | This will allow players to control actions and movements in video games by moving the device through the air. | This \|\| will \|\| allow \|\| players \|\| to \|\| con \|\| trol \|\| action \|\| s \|\| and \|\| movemen \|\| ts \|\| in \|\| video \|\| games \|\| by \|\| moving \|\| the \|\| device \|\| through \|\| the \|\| air. | This will \|\| allow players \|\| to control \|\| actions \|\| and move- ments \|\| in video games \|\| by \|\| moving \|\| the device \|\| through \|\| the air. | This \|\| will \|\| allow \|\| players \|\| to \|\| con \|\| trol \|\| actions \|\| and \|\| movemen \|\| ts \|\| in \|\| video \|\| games \|\| by \|\| moving \|\| the \|\| device \|\| through \|\| the \|\| air. | This \|\| will \|\| allow \|\| players \|\| to \|\| control \|\| actions \|\| and \|\| move- ments \|\| in \|\| video\|\| games\|\| by\|\| moving\|\| the \|\| device \|\| through \|\| the \|\| air \|\|. |
| es | Esto permitirá a los jugadores controlar las acciones y los movimientos en los videojuegos, a través del movimiento del dispositivo por el aire. | Es \|\| to \|\| permitir \|\| á \|\| a los \|\| jugadores \|\| con \|\| trolar \|\| las \|\| accion \|\| es \|\| y \|\| los \|\| movimien- tos \|\| en \|\| los \|\| videojuegos, \|\| a \|\| través \|\| del \|\| movimiento \|\| del \|\| dispositivo \|\| por \|\| el \|\| aire. | Esto permitirá \|\| a los \|\| jugadores \|\| controlar \|\| las acciones \|\| y los \|\| movimientos \|\| en los \|\| videojuegos \|\|, a través \|\| del movimiento \|\| del dispositivo por el aire. | Esto \|\| permitirá \|\| a \|\| los \|\| jugadores \|\| con \|\| trolar \|\| las \|\| acciones \|\| y \|\| los \|\| movimien \|\| tos \|\| en \|\| los \|\| videojuegos, \|\| a través \|\| de \|\| l movimien \|\| to \|\| del \|\| l dispositivo \|\| por \|\| e \|\| l aire. | Esto\|\| permitirá\|\| all los\|\| jugadores\|\| controlar\|\| las\|\| ac- ciones\|\| y\|\| los\|\| movimientos\|\| en\|\| los\|\| videojuegos,\|\| a través\|\| del\|\| l movimiento\|\| del\|\| dispositivo\|\| por\|\| el\|\| aire\|\|. |
| te | పరికరాన్ని గాల్లో కదిలించడం ద్వారా వీడియో గేమ్స్లో యాక్షన్లు మరియు కదలికలను నియంత్రించడానికి ఇది ఆటగాళ్లకు వీలు కల్పిస్తుంది. | _(segmented Telugu text)_ | _(segmented Telugu text)_ | _(segmented Telugu text)_ | _(segmented Telugu text)_ |
| hi | यह डिवाइस को हवा में मूव करके खिलाड़ियों को एक्शन और मूवमेंट कंट्रोल करने की अनुमति देगा. | _(segmented Hindi text)_ | _(segmented Hindi text)_ | यह डिवाइस क \|\| ो हवा म \|\| ें ज्व करके खिलाडिय \|\| ों क \|\| ो एक्शन और म \|\| ूवमेंट कंट \|\| ्रोल करन \|\| े की अन \|\| ुमति द \|\| ेगा. | _(segmented Hindi text)_ |
| ru | Посредством движения устройства в воздухе игроки смогут управлять действиями и движениями в видеоиграх. | П \|\| о \|\| сре \|\| дств \|\| о \|\| м \|\| дви \|\| жени \|\| я \|\| устро \|\| йств \|\| а \|\| в \|\| во \|\| зду- хе \|\| и \|\| гро \|\| ки \|\| смо \|\| гут \|\| упра \|\| вля \|\| ть \|\| де \|\| йств \|\| и \|\| ями \|\| и \|\| дви \|\| жени \|\| ями \|\| в \|\| ви \|\| део \|\| и \|\| гра \|\| х. | Посред \|\| ство \|\| м \|\| движен \|\| ия \|\| устрой ства \|\| в воздухе \|\| игро- ки \|\| смогут \|\| управлять \|\| действи \|\| ями \|\| и дви- жен \|\| иями \|\| в видеоиг- ра \|\| х | По \|\| средством \|\| дви- же \|\| ния \|\| устройства \|\| в воздухе \|\| и \|\| гроки \|\| смогут \|\| управлять \|\| дейст \|\| виями \|\| и \|\| дви- же \|\| ниями \|\| в видеои \|\| грах | Пос \|\| ред \|\| ством \|\| дви- жения \|\| устройства \|\| в \|\| воздухе \|\| игроки \|\| смо- гут \|\| управлять \|\| дей- ствия \|\| ми \|\| и \|\| движ \|\| ениями \|\| в\|\| видео \|\| иг \|\| рах \|\|. |
| uk | Це забезпечити грав- цям контроль над діями та рухами у відеоіграх, рухаючи пристрій у повітрі. | Подор \|\| ож н \|\| а мі \|\| сце \|\| можн \|\| а зр \|\| учно по \|\| єдна \|\| ти з пр \|\| огулян- кою озе \|\| ром \|\| у човні. | Це \|\| забезпечити \|\| грав- цям \|\| контро \|\| ль \|\| над \|\| діями \|\| та \|\| рухами \|\| у \|\| відеоігра \|\| х \|\|, рухаючи \|\| пристрій \|\| у повітрі. | Це \|\| забезпечити \|\| грав- цям \|\| контроль \|\| над \|\| ді- ми \|\| та рухами \|\| у відеоі- грах, \|\| рухаю \|\| чи \|\| при- стрій \|\| у повітрі. | Це \|\| забезпеч \|\|ить \|\| грав \|\|цям \|\| контроль \|\| над \|\| діями \|\| та \|\| рух \|\|ами \|\| у \|\| відео \|\|іг \|\|рах \|\|, \|\| рух \|\|аючи \|\| при- стрій \|\| у \|\| повітрі \|\|. |

## 5.3   Does segmentation significantly change after finetuning?

We investigate the effects of fine-tuning on the resulting segmentation boundaries across various downstream tasks. Essentially, we inspect how the boundary prediction changes after fine-tuning for each downstream task. We find that there are no differences in segmentation before and after fine-tuning despite updating the parameters of the boundary predictors. While there are a few instances where the segmentation of the fine-tuned model is different than that of the pretrained model, there is no clear evidence of the segmentation changing drastically after fine-tuning. In Table 7 in the Appendix, we present two examples from the SIB dataset where there is a slight change in segmentation. We found no indication that any observed changes contribute to or hurt task performance.

# 6 Related Work

**Overcoming segmentation disparities in subword tokenization**    In multilingual settings, subword tokenizers have proven to be prone to over-segmentation, due to the data-driven nature of the BPE algorithm [39]. Previous work [13, 12, 44] has attempted to address data imbalance issues in subword tokenizers by over-sampling low-resource languages. Our work shows that this only alleviates the bias on certain scripts and doesn't solve the problem. Other studies [1, 36, 17] have also shown that tokenization in transformers remains biased in favor of high-resource languages. Wang et al. [42] enforce models to use smaller subword units in high-resource languages to make segmentation fairer. Some works [23, 8] suggest training multilingual tokenizers on language clusters to mitigate segmentation disparities, however, this leads to expanded vocabularies. Despite these attempts, it is evident that the training objectives of subword tokenizers do not effectively align with those of language modeling.

**Tokenizer-free language models**    Language modelling over bytes [45, 4] and pixels [37, 38, 26] has become desirable, as it removes complicated preprocessing pipelines in modelling. Xue et al. [45] introduced ByT5, a tokenizer-free variant of T5 [35] that processes text at the byte level. However byte-level encoding over-fragments non-Latin script languages resulting in overly long sequences. Since byte or character sequences usually result in longer sequences, previous work [30, 9, 41, 47, 31, 15] on tokenizer-free LMs has introduced novel model architectures to mitigate the computational overhead of processing raw character or byte text directly. These methods [9, 41, 47, 30] end up segmenting raw sequences into fixed/dynamic-size patches, which is not suitable for modelling over non-Latin scripts.

# 7 Conclusion

In this work, we introduce MAGNET, a gradient-based tokenization method to learn equitable segmentation across languages scripts in byte-level multilingual models. MAGNET dynamically routes byte-level sequences through language-script-specific internal boundary predictors trained to infer word boundaries through stochastic reparameterisation. We show that MAGNET enables us to learn token representations with the same granularity across languages compared to vanilla byte-level models and previous gradient-based tokenization approaches. Our analysis demonstrates that while there are indeed downstream performance trade-offs as a result of MAGNET inducing high compression on non-Latin script languages, we are still able to maintain downstream performance quality. Overall, our results hold promise for future research on equitable segmentation and text processing more generally.

## Acknowledgments

We would like to thank the UW NLP community for valuable discussions of this work. We are grateful to Farhan Samir and Alisa Liu for discussions on experiments and analysis. This work was supported in part by NSF IIS 2113530. We also gratefully acknowledge support from the National Science Foundation under CAREER Grant No. IIS2142739, NSF grants No. IIS2125201 and IIS2203097, and gift funding from Google, MSR, and OpenAI.

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

# Appendix

## A    Limitations

The primary limitation of our work is the restricted resources available for the extensive experiments we carried out. This constrained the number of languages included in our pretraining data, the size of the pretraining data itself, and the size of the models. Nonetheless, we hypothesize that our current results will hold true when replicated with larger models, as MAGNET is likely to provide even greater benefits when integrated into large models. We leave this to future work to explore further. Another limitation is the performance-compression trade-offs associated with MAGNET, as some languages are sensitive to high compression rates. However we note that this is not universal across all tasks. In fact, we argue that MAGNET offers users the flexibility to optimize for their desired benefits. Finally, MAGNET also inherits some limitations from previous gradient-based tokenization methods [31, 30, 41] and the vast majority of segmentation methods. This approach may not be suitable for Semitic languages, where morphemes are discontinuous and vowels are interspersed between consonant roots for inflection or sometimes omitted. Computing byte-to-word ratios would be harder in CJK (Chinese, Japanese, and Korean) languages, as they do not mark word boundaries with space. However we note that our choice of languages was not influenced by space separation, but rather by linguistic diversity and the availability of data for downstream evaluation. MAGNET is very flexible and applicable to all languages that can be expressed with UTF-8, we use byte-word-ratio as a simple proxy to train our boundary predictors to learn equitable tokenization . We note that byte-word-ratio is not a compulsory proxy and for such languages other proxies can be used

## B    Broader Impacts Statement

In this work, we contribute to promoting equitable segmentation in multilingual language models across various language scripts. Our approach holds promise for enhancing the utility and efficiency of multilingual language models, particularly benefiting low-resourced and non-Latin script languages spoken by billions worldwide. We acknowledge limitations of our work in Appendix A, and strongly advise against unintended usage of the models. We will release our code and models to facilitate further research in this direction.

## C    Dataset Statistics

### C.1    Pretraining data

Our pre-training data is obtained from the OSCAR dataset [32]. Due to resource constraints, we only pretrain our models on a subset of this dataset. The distribution of tokens across languages in displayed in Figure 6.

### C.2    Downstream data

Table 5: Downstream language and task coverage

| Dataset | Task | Languages |
|---------|------|-----------|
| XNLI | Natural language inference | en, fr, es, ru, hi, bn , te |
| XQUAD | Question answering | en, es, ru, hi |
| PAWS-X | Adversarial paraphrase identification | en, fr, es |
| SIB-200 | Topic classification | en, fr, es, ru, uk, be, bn, te, hi |

## D    Technical Details

**Data Preprocessing**    For all our datasets, we preprocess all text to raw UTF-8 bytes. In the MAGNET models, we add a unique script identifier to the front of every sequence that guides the models to route the sequence to the respective script boundary predictor.

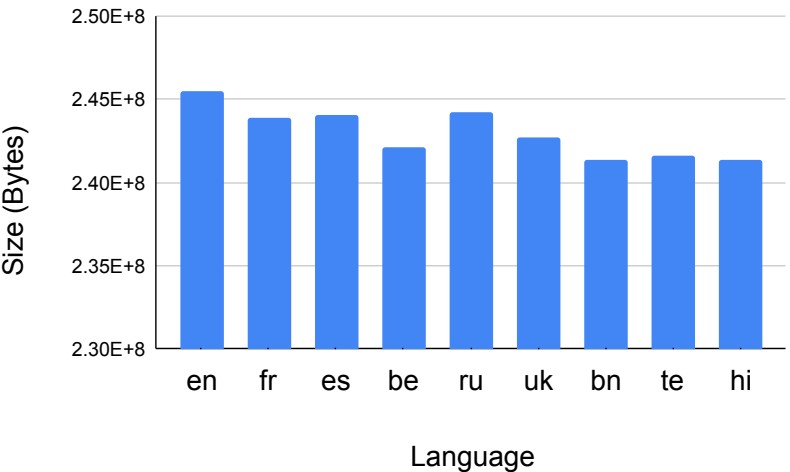

Figure 6: Language statistics in the pretraining data.

### D.1 Model Hyperparameters

For all our experiments, we use 14-layer hourglass transformers with 2 layers in the first block, 10 layers in the second block and 2 layers in the final block. For every transformer layer, the hidden dimension is 768, the intermediate feed-forward dimension is 3072. Each self-attention layer consists of 12 heads. We use a post-norm architecture, GELU activation function [18] in feedforward layers and the relative attention parametrisation from Transformer XL. This brings our model's size to ≈ 126M parameters. The boundary predictor is a 2-layer MLP that takes in a hidden state as input and outputs a scalar prediction at each time step. We use the Adam optimizer [21] with $(\beta_1, \beta_2)$ and $\epsilon$ parameters as (0.9, 0.98) and 1e-6, respectively. We use a learning rate of 5e-5, a warmup ratio of 0.1 and 38,000 training steps, a batch size of 512 distributed across 4 A40 GPUs. Each batch consists of examples concatenated up to the maximum sequence length of 512.

### D.2 Finetuning

We finetune the pretrained models by appending a linear classification layer and update all parameters during training. The boundary predictors' parameters are also updated to ensure that the predicted segmentations are well adapted to each task. We finetune for 5 epochs with a batch size of 32 and the same learning rate and warm-up ratio that we used for pretraining. We report accuracy averaged over 2 runs with different random seeds.

## E  Supplementary Results

### E.1 Equitable Segmentation at Byte-Level Granularity

In Figure 7, we present plots comparing segmentation granularity between (1.) byte-level segmentation and MAGNET$(1, 2, 4)\times$. (2) Between DTP $5\times$, MAGNET at $(5, 10, 13)\times$, MAGNET at $(5, 10, 20)\times$ and subword-tokenizers at vocabulary sizes 50k, 100k and 250k with and without alpha sampling (see section 3.1)

### E.2 Downstream Tasks

We present language-level results (accuracy) across all tasks and models in Table 6

Table 6: Language-level results across all tasks

| Model | en | fr | es | ru | hi | bn | te | Avg |
|---|---|---|---|---|---|---|---|---|
| Byte-Level | 74.72 | 70.89 | 71.77 | 67.89 | 62.89 | 67.93 | 66.90 | 69.12 |
| DTP $5\times$ | 73.18 | 70.19 | 69.78 | 66.96 | 61.79 | 67.11 | 66.12 | 66.74 |
| DTP $10\times$ | 71.92 | 68.05 | 68.13 | 67.00 | 61.26 | 65.79 | 65.49 | 66.38 |
| MAGNET$(1, 2, 4)$ x | 74.44 | 70.63 | 71.99 | 67.48 | 64.73 | 67.44 | 66.88 | 69.17 |
| MAGNET$(3, 6, 12)$ x | 74.41 | 70.83 | 70.78 | 67.66 | 61.57 | 65.32 | 65.14 | 67.53 |
| MAGNET$(5, 10, 20)$ x | 73.71 | 70.23 | 71.15 | 66.35 | 61.72 | 65.12 | 65.01 | 67.13 |
| MAGNET$(5, 10, 13)$ x | 73.97 | 70.38 | 70.58 | 66.85 | 62.42 | 66.11 | 65.46 | 67.97 |
| MAGNET$(5, 10, 15)$ x | 73.57 | 70.82 | 71.36 | 67.02 | 61.89 | 65.43 | 64.96 | 67.86 |

(a) XNLI

| Model | en | fr | es | Avg |
|---|---|---|---|---|
| Byte-Level | 85.70 | 80.45 | 80.40 | 82.18 |
| DTP $5\times$ | 84.13 | 80.20 | 79.55 | 81.29 |
| DTP $10\times$ | 76.13 | 76.10 | 75.75 | 75.99 |
| MAGNET$(1, 2, 4)\times$ | 87.63 | 81.73 | 80.55 | 83.30 |
| MAGNET$(3, 6, 12)\times$ | 87.60 | 83.98 | 84.65 | 85.41 |
| MAGNET$(5, 10, 20)\times$ | 82.05 | 78.83 | 80.45 | 80.44 |
| MAGNET$(5, 10, 13)\times$ | 81.40 | 78.28 | 79.10 | 79.59 |
| MAGNET$(5, 10, 15)\times$ | 87.78 | 83.05 | 83.55 | 84.79 |

(b) PAWSX

| | en | es | ru | hi | Avg |
|---|---|---|---|---|---|
| Byte-Level | 55.61 | 53.225 | 40.49 | 29.18 | 44.62 |
| DTP $5\times$ | 51.40 | 49.85 | 41.62 | 30.39 | 43.31 |
| DTP $10\times$ | 40.68 | 37.75 | 38.09 | 26.79 | 35.82 |
| MAGNET$(1, 2, 4)\times$ | 55.94 | 51.78 | 41.75 | 28.98 | 44.61 |
| MAGNET$(3, 6, 12)\times$ | 55.32 | 52.46 | 41.29 | 25.82 | 43.72 |
| MAGNET$(5, 10, 20)\times$ | 53.12 | 50.29 | 39.61 | 21.23 | 41.06 |
| MAGNET$(5, 10, 13)\times$ | 53.13 | 50.16 | 38.31 | 26.07 | 41.92 |
| MAGNET$(5, 10, 15)\times$ | 53.14 | 51.88 | 39.82 | 26.41 | 42.81 |

(c) XQUAD

| Model | en | es | fr | ru | be | uk | bn | te | hi | Avg |
|---|---|---|---|---|---|---|---|---|---|---|
| Byte-Level | 76.96 | 72.06 | 71.57 | 75.98 | 71.07 | 72.55 | 63.00 | 68.63 | 67.65 | 71.05 |
| DTP $5\times$ | 73.28 | 73.03 | 69.60 | 70.34 | 73.77 | 70.83 | 62.01 | 66.18 | 63.48 | 69.17 |
| DTP $10\times$ | 73.28 | 68.38 | 68.14 | 67.89 | 70.83 | 67.89 | 63.24 | 66.67 | 64.21 | 67.83 |
| MAGNET$(1, 2, 4)\times$ | 75.25 | 73.04 | 72.55 | 75.00 | 70.09 | 70.59 | 63.24 | 66.43 | 73.04 | 71.02 |
| MAGNET$(3, 6, 12)\times$ | 79.17 | 77.94 | 75.69 | 75.00 | 75.74 | 73.29 | 56.62 | 59.31 | 56.13 | 69.87 |
| MAGNET$(5, 10, 20)\times$ | 80.15 | 78.92 | 77.70 | 75.98 | 76.47 | 74.02 | 49.75 | 53.90 | 51.96 | 68.76 |
| MAGNET$(5, 10, 13)\times$ | 79.17 | 79.66 | 73.78 | 71.57 | 73.78 | 71.33 | 59.07 | 65.68 | 61.76 | 70.64 |
| MAGNET$(5, 10, 15)\times$ | 80.64 | 78.92 | 74.80 | 72.30 | 76.23 | 72.31 | 56.37 | 52.94 | 54.66 | 68.79 |

(d) SIB-200

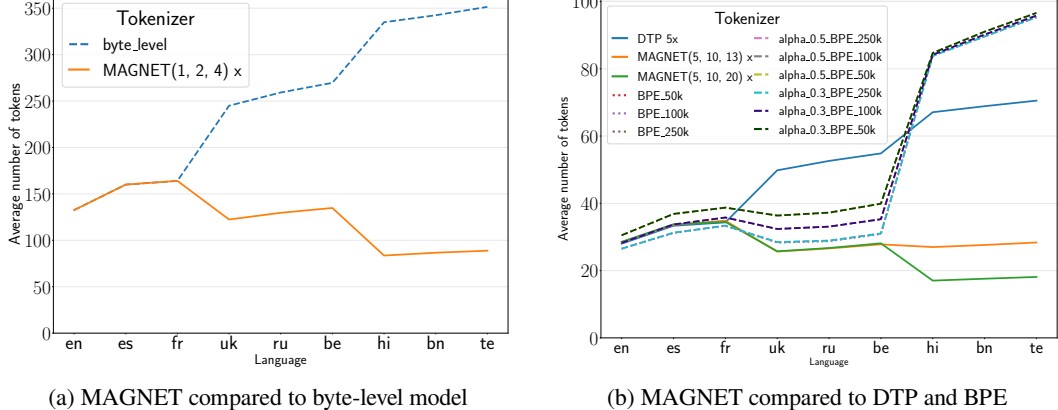

(a) MAGNET compared to byte-level model      (b) MAGNET compared to DTP and BPE

Figure 7: Average number of tokens after segmenting the FLORES dataset. Evidently subword tokenizers and DTP result in over-segmentation in non-Latin script languages, while MAGNET closes the gap.

Table 7: English and Russian instances from the SIB dataset showing slight changes in segmentation before and after fine-tuning. Segmentation for these examples was performed using the MAGNET (5x, 10x, 20x).

| Lang | Text | Finetuning Segmentation | Pretraining Segmentation |
|---|---|---|---|
| en | Last month a presidential commission recommended the prior CEP's resignation as part of a package of measures to move the country towards new elections. | Last ‖ month ‖ a ‖ presiden ‖ tial ‖ com-mission ‖ recommen ‖ ded ‖ the ‖ prior ‖ CEP's ‖ resign ‖ a ‖ tion ‖ as ‖ part ‖ of ‖ a ‖ package ‖ of ‖ measures ‖ to ‖ move ‖ the ‖ country ‖ towards ‖ new ‖ e ‖ lec ‖ tions. | Last ‖ mon ‖ th ‖ a ‖ presiden ‖ tial ‖ comm ‖ ission ‖ recommended ‖ the ‖ prior ‖ CEP's ‖ resign ‖ ation ‖ as ‖ part ‖ of ‖ a ‖ package ‖ of ‖ measures ‖ to ‖ move ‖ the ‖ country ‖ towards ‖ new ‖ e ‖ lections. |
| ru | В прошлом месяце президентская комиссия рекомендовала предыдущему Временному избирательному совету уйти в отставку в качестве части пакета мер для движения страны к новым выборам. | В про ‖ шлом ‖ месяце ‖ президе ‖ нтс ‖ кая ‖ коми ‖ сси ‖ я ‖ реко-ме ‖ ндовала ‖ предыд ‖ у ‖ щему ‖ Време ‖ нному ‖ избиратель ‖ ному ‖ совету ‖ уйти ‖ в отставку ‖ в ка-честве ‖ части ‖ пакета ‖ мер ‖ для движе ‖ ния ‖ страны ‖ к новым ‖ выборам | В про ‖ шлом ‖ месяце ‖ президе ‖ нтс ‖ кая ‖ комиссия ‖ рекоме ‖ ндо-вала ‖ предыду ‖ щему ‖ Време ‖ нному ‖ избиратель ‖ ному ‖ сове-ту ‖ уйти ‖ в отставку ‖ в качестве ‖ части ‖ пакета ‖ мер ‖ для движе ‖ ния ‖ страны ‖ к новым ‖ выборам. |

