# OpenReview forum: "MAGNET: Improving the Multilingual Fairness of Language Models with Adaptive Gradient-Based Tokenization"
_NeurIPS.cc/2024/Conference — NeurIPS 2024 poster_

### Official Review · Reviewer_2xYq · 2024-07-13

**Soundness:** 3
**Presentation:** 3
**Contribution:** 3
**Rating:** 6
**Confidence:** 3

**Summary:**

This paper proposed MAGNET, a gradient-based tokenization method, to address an over-segmentation issue when handling multilingual text data written in different scripts. MAGNET learns to predict segment boundaries between byte tokens in a text sequence. MAGNET has a customizable architecture where byte-level sequence are routed through language-script-specific predictors, which implements a language-dependent tokenization and thus avoids over-segmentation in non-Latin scripts. The authors conducted extensive experiments on nine languages with distinct scripts. Their results show that the proposed approach works well in the downstream task and moreover contributes to the speed-up at inference time.

This paper is well organized and clearly motivated. The over-segmentation gets a bigger issue in multilingual model training so the proposed approach will be useful.

**Strengths:**

- Well organized paper. easily to follow each section. enough description is provided.
- Extensive results with 9 languages with 3 different scripts.. the authors conducted extensive experiments in the downstream task
- Successfully reducing the inference time

**Weaknesses:**

- experiments are carried out in three different scripts. Other interesting languages would be Chinese, Japanese and Korean scripts as another major examples that struggle with the out-of-vocabulary issues.
- despite addressing the oversegmentation, performance is still similar to that of byte-level or baselines.

**Questions:**

- I see the proposed approach successfully overcoming segmentation disparities in subword tokenization though what else would be needed to improve the downstream task perfomance? For instance, Figure 3 shows that the byte-level model also achieves good scores across.

- Other interesting languages would be Chinese, Japanese and Korean scripts. Have you ever experimented with the languages? If yes, do you observe anything new?

---

> ### Author Rebuttal · Authors · 2024-08-07
>
> We thank reviewer 2xYq for taking the time to review our work and noting that our experiments and results are extensive with a successful reduction in inference time. We thank you for your suggestions and address your concerns below.
>
> **Sufficient language coverage**
>    - Our choice of languages was not influenced by space separation, but rather by linguistic diversity and the availability of data for downstream evaluation. Whilst we agree that CJK languages like Chinese and Japanese could have been included in our experiments, we were limited by computational resources and future work should extend our current work to these languages.
>    - MAGNET is very flexible and applicable to all languages that can be expressed with UTF-8, we use byte-word-ratio as a simple proxy to train our boundary predictors to learn equitable tokenization.  Indeed we cannot use byte-to-word ratio for languages without whitespace like CJK languages, but it's important to note that byte-word-ratio is not a compulsory proxy and for such languages other proxies can be used. Also computing the byte-to-word ratio is quite trivial and doesn’t require a huge amount of text. For CJK languages, one could employ state-of-the-art word-segmentation tools like Jieba to count the number of words before computing the byte-to-word ratio.
>
> **Despite addressing the over segmentation, performance is still similar to that of byte-level or baselines**
>    - The goal of MAGNET is not to outperform byte-level models or DTP, but to encourage multilingual language models to learn equitable segmentations that result in equitable model representations across languages.We still, however, see that MAGNET outperforms DTP on most tasks .
>
> **I see the proposed approach successfully overcoming segmentation disparities in subword tokenization though what else would be needed to improve the downstream task performance?**
>   - Due to computational limitations, we are restricted to pretraining on a smaller subset of data compared to state-of-the-art models. We believe that scaling the size of our models and pretraining data could significantly help improving downstream performance alongside overcoming segmentation disparities.

---

> > ### Comment · Reviewer_2xYq · 2024-08-13
> >
> > Thank you for your response. I have read it along with the rest of reviewers' comments and I have decided to keep my score unchanged. While I understand the authors' computational limitations in conducting experiments on CJK languages, though, as Reviewer stTy also noted, including experimental results on CJK languages would enhance technical soundness of the work.

---

### Official Review · Reviewer_KW7H · 2024-07-13

**Soundness:** 4
**Presentation:** 3
**Contribution:** 3
**Rating:** 7
**Confidence:** 3

**Summary:**

The authors propose MAGNET (multilingual adaptive gradient-based tokenization) to remove the common problem in non-Latin-script languages getting only tokens (subwords assigned in the vocabulary) representing short character sequences as opposed to English getting high-semantic-content tokens.

As opposed to previous gradient based tokenization strategies which globally minimize compression cost across all sequences, they use language specific segmentation predictors to reduce oversegmentation in non-Latin-script languages, to produce a better multilingual byte-level LM.

Within a byte-level LM the tokenization acts as an information bottleneck, where the input subword byte sequences have to be predicted again at the output using an "upsampling module", while next word prediction takes place over the tokenized-on-the-fly blocks.

They compare MAGNET to DTP (which doesn't have the same conditioning but otherwise has identical structure) and naive BPE over the vocabulary. Figure 2 shows that the avg # tokens per passage across all languages, as opposed to BPE's severe jump for Hindi, Bengali, and Telegu, and DTP's punishment of both Cyrillic and Indic script languages.

**Strengths:**

This tokenization issue comes up a lot in multilingual NLP research. Different segmentation levels between languages can severely hamper LM performance---particularly open source, open weight LM performance---on many cross-lingual tasks including summarization, translation, QA, and reasoning, so fixing this with architectures such as this and scaling them will prove very impactful.

This simple proposal to fit the hourglass architecture to script-denominated segementation predictors is a simple but useful refinement.

Simple presentation of their method's superior results for simple tokenization, which also generalized to better performance on most multilingual tasks such as XNLI, PAWS-X, and SIB.

The savings in token cost for passages translate into better inference time as well.

**Weaknesses:**

Language ID is still an issue here. While they are able to fit these individual segmentation predictors based on implicit language family by script (which can be inferred from position in the UTF-8 table), I think the case could be made that they're basically just kicking the unequal tokenization can down a level. Now words in Telegu might be worse segmented than words in Hindi, etc. Really an inline language identification-conditioned segmentation module would be best here. But that's probably better as a direction for future work anyway.

**Questions:**

N/A

**Limitations:**

Yes, they mention language-specific issues to expanding their method beyond the alphabetic languages they examined. I would like to see them mention the further potential improvements to using inline language ID prediction (conditioned in training, maybe predicted implicitly in inference?) to control the segmentation module and maybe ensure more equitable performance across the languages.

---

> ### Author Rebuttal · Authors · 2024-08-07
>
> We thank reviewer KW7H for reviewing our work and noting that MAGNET is very impactful and useful in reducing over-segmentation in multilingual language models. They also note that MAGNET results in better performance and lower inference costs. We appreciate your suggestions and address your concerns below.
>
> **Resolving Language ID issues**
>  - Yes, this is a valid point and a great suggestion. Currently, the number of boundary predictors in MAGNET can be scaled based on the linguistic properties that the languages in the pretraining data share. This is straightforward given that the boundary predictor is a small module within the entire model.  However, a language identification-conditioned segmentation module that doesn’t require adding too many individual boundary predictors could also be relevant here, and this would be a great avenue for future work. We will include this discussion in the limitations and future work section of the final version.

---

> > ### Comment · Reviewer_KW7H · 2024-08-13
> >
> > Thanks for your response!

---

### Official Review · Reviewer_WSTY · 2024-07-14

**Soundness:** 3
**Presentation:** 3
**Contribution:** 2
**Rating:** 6
**Confidence:** 4

**Summary:**

In this paper, the authors propose a multilingual adaptive gradient-based tokenization approach to reduce over-segmentation for non-Latin language texts. In particular, they improve the previous Dynamic Token Pooling method by inferring token boundaries with different predictors for different languages. They conduct extensive experiments to demonstrate that not only the over-segmentation issue is reduced, the inference efficiency also increases.

**Strengths:**

* Apply different boundary predictors for different languages/scripts, which is a natural improvement over the previous approach, DTP
* The results look reasonable and similar to human tokenization

**Weaknesses:**

* Basic token prediction tasks or text generation tasks are not included in the evaluation
* Downstream tasks' performances aren't different
* Inference time improvement isn't much compared to DTP

**Questions:**

In Figure 2, are some lines missing or overlapped?

**Limitations:**

Yes

---

> ### Author Rebuttal · Authors · 2024-08-07
>
> We thank reviewer WSTY for taking time to review our work and noting that our improvements over previous work yield better tokenization. We address your concerns below.
>
> **Basic token prediction tasks or text generation tasks are not included in the evaluation**
>   - We limited our evaluation to text understanding tasks following lots of prior work on tokenization Godey et al. (2022),  Clark et al. (2022), Tay et al. (2022). We believe that future work should focus on text-generation tasks. Text generation tasks are indeed byte-level and slower, and follow-up ideas similar to Fleshman et al. (2023)  can address them.
>
> **Downstream tasks' performances aren't different**
>   - The goal of Magnet is not to outperform byte-level models or DTP, but to encourage multilingual language models to learn equitable segmentations that result in equitable model representations across languages. However, we still see that Magnet outperforms DTP on most tasks.
>
> **Inference time improvement isn't much compared to DTP**
>  - Inference time improvements compared to DTP do not appear significant due to some in-effciencies in our implementation. We will invest more time into making the implementation more efficient for the final version of the paper.
>
> &nbsp;
> &nbsp;
> &nbsp;
>
> Godey, Nathan, et al. "MANTa: Efficient Gradient-Based Tokenization for Robust End-to-End Language Modeling." EMNLP 2022-The 2022 Conference on Empirical Methods in Natural Language Processing. 2022
>
> Clark, Jonathan H., et al. "Canine: Pre-training an Efficient Tokenization-Free Encoder for Language Representation." Transactions of the Association for Computational Linguistics 10 (2022)
>
> Fleshman, William, and Benjamin Van Durme. "Toucan: Token-Aware Character Level Language Modeling." arXiv preprint arXiv:2311.08620 (2023)

---

### Official Review · Reviewer_stTy · 2024-07-16

**Soundness:** 3
**Presentation:** 3
**Contribution:** 2
**Rating:** 5
**Confidence:** 3

**Summary:**

This paper presents multilingual adaptive gradient-based tokenization (MAGNET), which aims to reduce over-segmentation in non-Latin script languages in multilingual settings. MAGNET processes byte-level sequences and routes them through language-script-specific predictors, each optimized for its respective script, such as Latin, Cyrillic, and Indic. The segmentation is modeled using a sigmoid function, making it differentiable. Experimental results show that MAGNET can maintain downstream task performance while reducing inference latency.

**Strengths:**

1. The main contribution of MAGNET is its ability to maintain performance on downstream tasks while improving inference time by reducing tokenized sequence length. Compared with byte-level tokenizers, inference is more than twice as fast while maintaining or even improving performance.
2. The idea is simple and straightforward, making the paper easy to follow.

**Weaknesses:**

1. The proposed method doesn't conduct experiments on languages with sufficient coverage. The predictors only include Latin, Cyrillic, and Indic, raising concerns about its applicability to languages lacking spaces as word boundaries, such as Chinese and Japanese.
2. The importance of byte-level tokenization and gradient-based tokenizers as research directions is unclear. The experimental results don't demonstrate the significance of byte-level tokenization in terms of downstream performance and inference latency.
3. The training objective remains byte-level without segmentation. This raises questions about whether MAGNET still needs to generate very long byte sequences during inference.

**Questions:**

1. Is there any analysis of the segmentation? For example, can the segmentation find phrase boundaries?
2. Is it possible to incorporate prior domain knowledge (such as word dictionaries or tokenization from other tokenizers) to improve gradient-based tokenization?
3. Are these three predictors sufficient to achieve good performance across all languages? What's the performance for languages that don't have spaces in the sequence, such as Chinese or Japanese?

**Limitations:**

Please refer to weaknesses.

---

> ### Author Rebuttal · Authors · 2024-08-07
>
> We thank reviewer stTy for taking the time to review our work and noting that MAGNET maintains performance on downstream tasks while improving efficiency and combating over-segmentation. We address your concerns below.
>
> **Sufficient language coverage**
>   - Our choice of languages was not influenced by space separation, but rather by linguistic diversity and the availability of data for downstream evaluation. Whilst we agree that CJK languages like Chinese and Japanese could have been included in our experiments, we were limited by computational resources and future work should extend our current work to these languages.
>  - MAGNET is very flexible and applicable to all languages that can be expressed with UTF-8, we use byte-word-ratio as a simple proxy to train our boundary predictors to learn equitable tokenization.  Indeed we cannot use byte-to-word ratio for languages without whitespace like CJK languages, but it's important to note that byte-word-ratio is not a compulsory proxy and for such languages other proxies can be used. Also computing the byte-to-word ratio is quite trivial and doesn’t require a huge amount of text . For CJK languages, one could employ state-of-the-art word-segmentation tools like Jieba to count the number of words before computing the byte-to-word ratio.
>
> **Byte-level and gradient-based tokenization research directions**
>  - Several works Ahia et al. (2023), Petrov et al. (2023) have pointed out flaws of subword level tokenization algorithms, particularly over-segmentation in non-Latin script languages. Tokenization in general is an active research area and there has been recent efforts to make it more robust and easily adaptable across languages and data domains Clark et al. (2022), Xue et al. (2022), Tay et al. (2022), Yu et al. (2023). Byte-level models are very relevant because of their high coverage since UTF-8 supports most of the world’s scripts. They have also generally been shown to match or surpass subword-models in performance. Moreover, gradient-based tokenization makes byte/character level models fairer and more efficient. We will highlight this better in our contributions in the final draft.
>
> **Training objective**
>  - The training objective is indeed byte-level and slower, and follow-up ideas similar to Fleshman et al. (2023)  can address them. For this reason, we limited our evaluation to text understanding tasks following lots of prior work on tokenization Godey et al. (2022), Clark et al. (2022),Tay et al. (2022) . We believe that future work should focus on text-generation tasks.
>
> **Is there any analysis of the segmentation? For example, can the segmentation find phrase boundaries?**
>   - MAGNET is customizable and the boundary predictors can be trained to learn different granularities of segmentations including phrase segmentations. We provide qualitative analysis of the segmentations learned by MAGNET in comparison to DTP and subword-level models in Table 5 and 6 in the Appendix.
>
> **Is it possible to incorporate prior domain knowledge (such as word dictionaries or tokenization from other tokenizers) to improve gradient-based tokenization?**
>   - Yes, this is very much possible. Although we train our boundary predictors via stochastic reparameterization, we note that one can also incorporate supervision from other sources such as word dictionaries or even BPE tokenizers. DTP Nawrot et al. (2023) experimented with this in their work.
>
> **Are these three predictors sufficient to achieve good performance across all languages?**
>   - We used three predictors in our experiments because the languages we covered belonged to three language subfamilies. First, the number of boundary predictions could be scaled based on user preference. Second, the trained boundary predictors are sufficient to cover languages with properties similar to those in the training data. Beyond languages in our pre-training data, we also conducted downstream experiments on 5 other Indo-Aryan languages. We observed that the predicted segmentations on these languages are close to the word-level segmentations we see in language.
>
> &nbsp;
> &nbsp;
> &nbsp;
> ---
> Petrov, Aleksandar, et al. "Language model tokenizers introduce unfairness between languages." Advances in Neural Information Processing Systems (2023)
>
> Ahia, Orevaoghene, et al. "Do All Languages Cost the Same? Tokenization in the Era of Commercial Language Models." Proceedings of the 2023 Conference on Empirical Methods in Natural Language Processing. 2023.
>
> Clark, Jonathan H., et al. "Canine: Pre-training an Efficient Tokenization-Free Encoder for Language Representation." Transactions of the Association for Computational Linguistics 10 (2022)
>
> Xue, Linting, et al. "Byt5: Towards a token-free future with pre-trained byte-to-byte models." Transactions of the Association for Computational Linguistics 10 (2022)
>
> Tay, Yi, et al. "Charformer: Fast Character Transformers via Gradient-based Subword Tokenization." International Conference on Learning Representations.
>
> Yu, Lili, et al. "Megabyte: Predicting million-byte sequences with multiscale transformers." Advances in Neural Information Processing Systems 36 (2023)
>
> Fleshman, William, and Benjamin Van Durme. "Toucan: Token-Aware Character Level Language Modeling." arXiv preprint arXiv:2311.08620 (2023)
>
> Godey, Nathan, et al. "MANTa: Efficient Gradient-Based Tokenization for Robust End-to-End Language Modeling." EMNLP 2022-The 2022 Conference on Empirical Methods in Natural Language Processing. 2022
>
> Nawrot, Piotr, et al. "Efficient Transformers with Dynamic Token Pooling." Proceedings of the 61st Annual Meeting of the Association for Computational Linguistics (Volume 1: Long Papers). 2023

---

> ### Comment · Reviewer_KW7H · 2024-08-13
> **Unclear on the importance of your weaknesses**
>
> I think it's worth reconsidering the first two weaknesses, as the authors note.
>
> 1. Showing improvement on a meaningful set of languages is a useful contribution, even if it isn't applied to a comprehensive set of popular languages. Additionally, CJK languages are much more well-studied than Indic languages as is; and they also have more information content per byte so segmentation is less of an issue than for alphabetic languages. The authors might want to note this
> 2. Byte-level tokenization is important in a lot of production, non-generic LLM language-model based applications such as ASR retrieval and others. The authors also note a lot of recent related work. Improving it is an important direction.
>
> Hope you consider these when responding to the authors and considering your score!

---

### Decision · Program_Chairs · 2024-09-25

**Decision:**

Accept (poster)

**Comment:**

The paper tackles a well-known issue in multilingual NLP, where tokenization discrepancies lead to unfair performance across languages. Reviewer KW7H highlights the importance of this: “This tokenization issue comes up a lot in multilingual NLP research. Different segmentation levels between languages can severely hamper LM performance...so fixing this with architectures such as this and scaling them will prove very impactful.” It does so with a simple yet novel approach, MAGNET, which introduces a customizable, gradient-based tokenization method that adapts to different language scripts. The experiments are also strong, as the paper demonstrates nine languages that MAGNET reduces over-segmentation and improves inference time on while maintaining or exceeding downstream task performance compared to baselines. Reviewer 2xYq acknowledges the thoroughness: “Extensive results with 9 languages with 3 different scripts.. the authors conducted extensive experiments in the downstream task.” There were some concerns about the limited language coverage, particularly the absence of CJK languages. I agree with 2xYq and stTy that including them would enhance the technical soundedness of this work, but accept the authors' rationale regarding this issue, "Our choice of languages was not influenced by space separation, but rather by linguistic diversity and the availability of data for downstream evaluation. Whilst we agree that CJK languages like Chinese and Japanese could have been included in our experiments, we were limited by computational resources". Also KM7H makes a good point that "CJK languages are much more well-studied than Indic languages as is; and they also have more information content per byte so segmentation is less of an issue than for alphabetic languages."

Overall the paper received positive reviews, and most concerns were addressed satisfactorily during the rebuttal in my opinion, so I vote to accept the paper.